# Hydrogen bonds to Au atoms in coordinated gold clusters

Md. Abu Bakar[1], Mizuho Sugiuchi[2], Mitsuhiro Iwasaki[2], Yukatsu Shichibu[1,2] & Katsuaki Konishi [1,2]

It is well known that various transition elements can form M⋯H hydrogen bonds. However, for gold, there has been limited decisive experimental evidence of such attractive interactions. Herein we demonstrate an example of spectroscopically identified hydrogen bonding interaction of C–H units to Au atoms in divalent hexagold clusters ($[Au_6]^{2+}$) decorated by diphosphine ligands. X-ray crystallography reveals substantially short Au–H/Au–C distances to indicate the presence of attractive interactions involving unfunctionalized C–H moieties. Solution $^1H$ and $^{13}C$ NMR signals of the C–H units appear at considerably downfield regions, indicating the hydrogen-bond character of the interactions. The Au⋯H interactions are critically involved in the ligand-cluster interactions to affect the stability of the cluster framework. This work demonstrates the uniqueness and potential of partially oxidised Au cluster moieties to participate in non-covalent interaction with various organic functionalities, which would expand the scope of gold clusters.

---

[1] Faculty of Environmental Earth Science, Hokkaido University, Sapporo, Hokkaido 060-0810, Japan. [2] Graduate School of Environmental Science, Hokkaido University, Sapporo, Hokkaido 060-0810, Japan. Correspondence and requests for materials should be addressed to K.K. (email: konishi@ees.hokudai.ac.jp)

For several decades, interatomic forces between metal (M) and hydrogen atoms (M···H or H···M) have attracted continuing interest not only from the fundamental aspects of chemical bonding but also in relation to their involvement in some organometallic catalysts. One of the typical examples is the "agostic" bond used for 3-center-2-electron C–H···M systems[1–4]. On the other hand, similar but different types of interactions that cannot be categorized in the agostic family are also known[1, 5–9]. Various terms, e.g., anagostic and preagostic, have been proposed for the description of such M···H–C systems, but they are virtually electrostatic-based attractive forces and are more similar to hydrogen bonds. In this relation, numerous examples of M···H "hydrogen bonds" involving O–H/N–H donor groups have been reported for the complexes of various transition metal elements[5, 10–18].

Among late-transition metal elements gold occupies a special position because of the strong relativistic effect[19]. For the interaction with hydrogen atoms, plenty of examples of close contacts with hydrogen atoms have been reported in the crystal structures of Au⁻, Au⁺, and Au³⁺ complexes[20]. However, as claimed in a recent review by Schmidbaur et al., the reported contacts are mostly due to the ligand-counterion interaction and/or crystal packing rather than the attractive interactions between Au and H atoms. Very recently, the first example of "agostic" C–H···Au interactions was reported[21], but even now there have been no examples of spectroscopically identified "hydrogen-bond type" Au···H interactions[22]. This is contrasted with the cases of the other transition metal complexes (e.g., platinum[11–16]), which offer abundant examples of M···H hydrogen bonds.

Ligand-protected gold clusters have currently attracted attention as a class of molecule-like metal species residing between particles and simple complexes[23–26]. Recent steep advances in the experimental/theoretical structural studies have revealed the critical involvement of the residual 6 s electrons in the emergence of unique structural and optical/electronic features, which lead to the development of superatom concepts. One of the emerging topics in this research area is unique catalysis[27–30]. However, the mechanism behind the catalytic processes, e.g., the interaction/activation at the gold surface, is still ambiguous because of the lack of information on the nature of interaction of the gold core with organic substrates.

In this work, we provide an example of Au···H–C hydrogen bonds in diphosphine-ligated divalent Au₆ clusters, which are firmly evidenced by both X-ray crystallography and solution nuclear magnetic resonance (NMR) spectroscopy, and reveal that hydrogen bonding to Au atoms is a possible interaction mode between organic moieties and an Au cluster. We also demonstrate

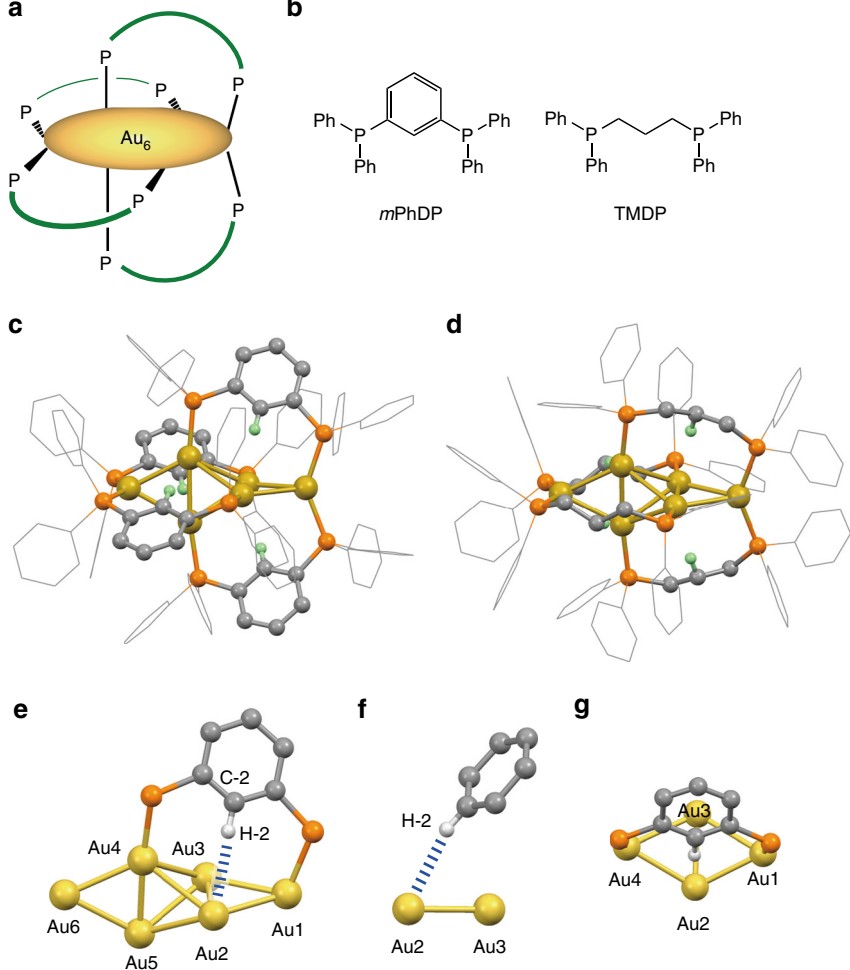

**Fig. 1** Structures of the diphosphine ligands and the hexagold clusters. **a** Schematic illustration of the Au₆ cluster decorated by four diphosphine ligands. **b** Chemical structures of the *m*PhDP and TMDP ligands. **c, d** X-ray crystal structure of the cationic moieties of **1**·(PF₆)₂ (**c**) and **2**·(NO₃)₂ (**d**). H-2 atoms nearest neighbour to the cluster moieties highlighted in *light green* with the other H atoms omitted for clarity. Au, P, and C atoms are coloured in *yellow*, *orange*, and *gray*, respectively. **e–g** Partial structures of **1**. **e** A bird view showing the Au₆ unit and a diphosphine ligand moiety. **f** A side view highlighting the Au···H–C interaction. **g** A top view showing the phenylene unit and neighbouring four Au atoms

**Table 1 Distances of H-2 and C-2 to the nearest-neighbor Au atoms and the corresponding C–H–Au angles in the crystal structures of 1·(PF$_6$)$_2$ and 2·(NO$_3$)$_2$**

| Cluster | Distances (Å) | | C–H–Au |
|---|---|---|---|
| | Au–H[a] | Au–C | Angle (°)[a] |
| **1**·(PF$_6$)$_2$ | 2.65 (2.777) | 3.699 | 163.0 (163.79) |
| | 2.60 (2.723) | 3.641 | 162.0 (162.82) |
| | 2.62 (2.748) | 3.692 | 171.0 (171.46) |
| | 2.60 (2.722) | 3.652 | 165.8 (166.43) |
| Ave. | 2.62 (2.743) | 3.671 | 165.4 (166.13) |
| **2**·(NO$_3$)$_2$ | 2.71 (2.821)[b] | 3.810 | 176.8 (175.83) |
| | 2.80 (2.914)[b] | 3.900 | 175.1 (173.67) |
| | 2.74 (2.853)[b] | 3.839 | 175.8 (174.54) |
| | 2.87 (2.978)[b] | 3.967 | 178.0 (177.43) |
| Ave. | 2.78 (2.892)[b] | 3.879 | 176.4 (175.37) |
| van der Waals contacts | 2.86[c] | 3.94-3.96[c] | 180 |

[a]Corrected distances and angles by using standard C–H bond lengths (1.08 (C($sp^2$)-H) and 1.10 Å (C($sp^3$)-H)). Observed distances in the X-ray crystal structures are shown in parentheses
[b]For one of the two CH$_2$ hydrogen atoms with shorter distances
[c]Estimated based upon standard van der Waals radii (H, 1.20 Å; C, 1.70 Å; Au, 1.66 Å) and C–H bond lengths with the assumption of linear arrangement of C, H, and Au atoms

the electronic coupling of the Au$_6$ cluster unit with neighbouring π-systems, and possible contributions of the Au···H–C interactions to the maintenance of the cluster framework to enhance the intrinsic stability.

## Results

**Synthesis and crystal structures of Au$_6$ clusters.** The gold clusters we employed in this study here have a core + exo-type Au$_6$ framework decorated by four diphosphine ligands (Fig. 1a). During the course of studies exploring new diphosphine-ligated Au clusters[25], we preliminarily found that the solution colours of two Au$_6$ clusters decorated by m-phenylene-bridged ([Au$_6$(mPhDP)$_4$]$^{2+}$, **1**) and by trimethylene-bridged diphosphines ([Au$_6$(TMDP)$_4$]$^{2+}$, **2**) (Fig. 1a, b) are significantly different. Since the electronic structure features and optical properties of molecular-sized Au clusters are essentially governed by their nuclearity and geometrical structures of the metallic units[31, 32], the above significant ligand effect is puzzling, which motivated us to obtain further insights from structural aspects.

The cluster carrying m-phenylene-bridged diphosphines (**1**) was synthesized in a similar manner to that reported previously for the synthesis of a trimethylene-bridged cluster (**2**)[33, 34]. Briefly, the treatment of [Au$_9$(PPh$_3$)$_8$](NO$_3$)$_3$ with mPhDP (Fig. 1b) in dichloromethane resulted in an instant color change from yellowish brown to greenish blue. The cationic cluster generated was isolated as the PF$_6$ salt, which was thoroughly characterized by means of mass spectrometry, elemental analysis, and single-crystal X-ray analysis. For example, electrospray ionization mass (ESI-MS) spectrum in methanol showed a sole set of signals at ~1484 in the range 1–6 kD, which was unambiguously assigned to the divalent cluster cation through comparison with the simulated isotope distribution pattern (Supplementary Fig. 1). X-ray crystallographic analysis (Fig. 1c) revealed that the geometrical structure of the Au$_6$ unit is very similar to that of **2** (Fig. 1d), having a core + exo type structure composed of a tetrahedral Au$_4$ core and two attached gold atoms at the exo positions. For instance, the Au–Au distances of **1** fell in the range 2.624–2.929 Å, which is close to that of **2** (2.614–2.966 Å).

The crystal structure also revealed close contacts of the gold framework to the bridging m-phenylene units. As shown in the partial structures (Fig. 1e–g), the hydrogen atom at the 2-position of an m-phenylene bridge (H-2) is located in proximity to Au2 with an apparent distance of 2.723 Å. For these results, one has to consider that the H positions were determined by the calculation based on the geometrically determined positions (riding model), giving shorter C–H bond lengths (~0.95 Å) than the "true" C–H lengths (1.08 Å)[35]. By simple geometrical calculation, the distance between H-2 and Au2 (Fig. 1c) was corrected to be 2.60 Å (Table 1), which is definitively shorter than the sum of the van der Waals radii (2.86 Å). Accordingly, the distance between C-2 and Au2 atoms in Fig. 1c (3.641 Å) was explicitly shorter than the Au–C distance when C, H, and Au atoms are aligned with van der Waals Au–H contact (3.95 Å). As summarized in Table 1, the other three phenylene units also showed sufficiently short Au–H and Au–C distances, which fall in the ranges of 2.60–2.65 Å (average 2.62 Å) and 3.641–3.699 Å (average 3.671 Å), respectively. Therefore, all four phenylene bridges are clipped to the Au$_6$ unit by attractive interactions. The Au–H–C angles for the above short Au–H contacts were in the range of 162.0–171.0° (Table 1). Therefore, considering the criterion for the distinction between the agostic/anagostic interactions[1], the interactions between the Au and H atoms should have hydrogen-bond characters rather than agostic.

Similar trends were observed in the crystal structure of the trimethylene-bridged Au$_6$ cluster (**2**), where short Au–H and Au–C distances were found for a hydrogen atom of the central CH$_2$ units (Fig. 1d). The corrected Au–H distances were 2.71–2.87 Å (average 2.78 Å) (Table 1), which are longer than those observed in **1** (2.60–2.65 Å) but still shorter than the sum of van der Waals radii (2.86 Å). Since the C–H–Au angles for the shortest contacts were all near 180° (Table 1) similarly to the conventional hydrogen bonds, it is suggested that hydrogen-bond-like Au···H–C interactions also exist in **2**. Nevertheless, the Au···H–C interactions appear weak when compared with those observed for the m-phenylene analogue (**1**), considering substantial differences in the Au–H and Au–C distances.

**Solution NMR spectra.** Based upon the above observations, we next investigated the solution structures by NMR spectroscopy. In dichloromethane-d$_2$, the four diphosphine ligands of **1**·(PF$_6$)$_2$ were NMR equivalent to give a single set of signals. For example, the $^{31}$P NMR spectrum exhibited only two peaks with comparable intensities (Supplementary Fig. 2), in agreement with the symmetrical feature of the X-ray structure. $^1$H and $^{13}$C NMR spectra, which were assigned with the aid of the $^1$H-$^1$H COSY and HSQC spectra (Supplementary Figs. 3–5), showed quite distinctive features for the signals due to the bridged m-phenylene moieties. In the $^1$H NMR spectrum, a signal was observed at an exceptionally high frequency (downfield) region (δ 11.6) (Fig. 2a), which was unambiguously assigned to H-2 of the m-phenylene bridges, while the other phenylene (H-4–H-6) and P-Ph protons gave signals at the normal aromatic region (δ 6.5–8). The effect of the counter anion was negligible since the spectra of **1**·(PF$_6$)$_2$ and **1**·(NO$_3$)$_2$ were almost identical to each other (see the data given in the Methods section and Supplementary Fig. 6). On the other hand, in the spectrum of the digold(I) mPhDP complex ((Au$_2$(mPhDP)Cl$_2$, **3**), all aromatic proton signals were found in the normal region (Fig. 2b and Supplementary Fig. 7). When this dinuclear complex (**3**) was used as a reference, the H-2 signal of **1** underwent a downfield shift of more than 4 ppm, while the other phenylene protons (H-4–H-6) shifted upfield by as large as 1 ppm (Table 2).

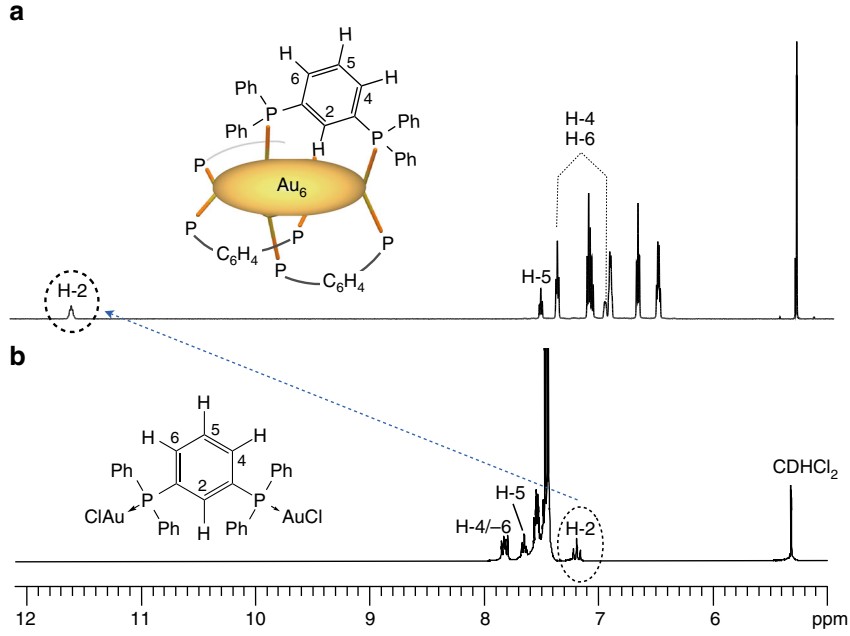

**Fig. 2** Comparison of the Au$_6$ cluster and the reference complex. **a**, **b** $^1$H NMR spectra of **1**·(PF$_6$)$_2$ (**a**) and **3** (**b**) in CD$_2$Cl$_2$ at 24 °C. The assignments for the phenylene-bridge protons are shown. The *arrow* indicates the downfield shift of the H-2 proton signals. The other unlabeled signals are due to the P-Ph protons

**Table 2 Chemical shifts for the Au$_6$ clusters (1 and 2) and the reference dinuclear complexes (3 and 4)**

| Proton | **1**·(PF$_6$)$_2$ | **3** | Δδ |
|---|---|---|---|
| H-2 | 11.57 | 7.20 | 4.37 |
| H-4, -6 | 6.97, 7.38 | 7.82 | −0.85, −0.44 |
| H-5 | 7.52 | 7.64 | −0.12 |
| | **2**·NO$_3$ | **4** | |
| H-2 | 3.05 | 1.92 | 1.13 |
| H-1, 3 | 2.08, 2.23 | 2.79 | −0.69, −0.56 |

In CD$_2$Cl$_2$ at 24 °C. Chemical shifts (δ ppm) were determined with respect to the residual solvent signals (δ 5.32)

It is generally known that the formation of hydrogen-bond type M···H interactions results in a downfield shift of the NMR resonance similarly to conventional hydrogen bonds[1, 5, 9]. For the difference of the chemical shifts between **1** and **3**, one of the possible factors is the magnetic shielding/deshielding effects of neighbouring P-Ph rings. Taking together the X-ray structure (Fig. 1c), the upfield shifts for the H-4 and H-6 signals, which were <0.8 ppm (Table 2), can be reasonably ascribed to the shielding effects of the Ph rings. On the other hand, the downfield shift of H-2 was much more explicit (>4 ppm). Such a large shift may not be explained only with deshielding effect of the neighbouring P-Ph rings. Therefore, it is likely that the downfield signal of H-2 is mainly due to the hydrogen-bond type "anagostic" interaction with the Au$_6$ cluster. It is known that "agostic" interactions generally offer low coupling constants $^1J_{CH}$ (50–100 Hz) for the C–H units when compared with the free C–H units[1]. In the present case, the $^1J_{CH}$ for the H-2 and C-2, estimated by the HMBC spectra, was 164 Hz, which is similar to the value of benzene C–H (159 Hz), thus indicating the "anagostic" character of the interaction of H-2 with the cluster. The involvement of the C–H moieties in the hydrogen-bond type interaction was further supported by $^{13}$C NMR profiles. The carbon atoms at the

2-position of the *m*-phenylene bridges (C-2) showed signals considerably downfield (δ 147.3) from the other aromatic carbons (δ 129–137) (Supplementary Fig. 4).

As mentioned, the four phenylene units gave a single set of signals in NMR. Thus they equally interact with the Au$_6$ unit on the NMR time scale. On the other hand, in the X-ray structure, each of the H-2 atoms interacts with single specific Au atoms (e.g., Au2, Fig. 1e–g), while the neighbouring Au atoms (e.g., Au3) appear to have no substantial interactions (Au–H distances: 3.02 Å in average). If such structures are retained in solution, the signals due to the P-Ph moieties should be resolved. However, no splitting behaviours were observed even at low temperature (−60 °C). Therefore, Au2 and Au3 may share the binding sites for H-2 as a result of the fast oscillation (flipping) of the phenylene rings. In this relation, it is generally known that the valence electrons of ligand-protected clusters are delocalized over the whole metallic units[24, 31]. Accordingly, the four-electron [Au$_6$]$^{2+}$ system appears to serve like an electron pool, facilitating the electrostatic interactions with neighbouring four C(δ-)-H(δ+) units. The present finding implies the inherent activity of partially oxidised gold units to serve as a "soft" hydrogen-bond acceptor (Lewis base).

We also measured the $^1$H NMR spectra of the trimethylene-bridged Au$_6$ cluster (**2**·(NO$_3$)$_2$) and the dinuclear reference complex (Au$_2$(TMDP)Cl$_2$, **4**) (Supplementary Fig. 8), and found a similar trend. As summarized in Table 2, the terminal CH$_2$ units (H-1 and H-3) of **2**·(NO$_3$)$_2$ gave signals at δ 2.08 and 2.23, which are shifted upfield from **4** by 0.69 and 0.56 ppm, respectively. These upfield shifts fall in the range observed for the H-4 and H-6 of **1**·(PF$_6$)$_2$ (0.44 and 0.85 ppm). The central CH$_2$ (H-2) showed a downfield shift similarly to H-2 of **1**, but the degree was much less prominent (1.13 vs. 4.37 ppm for **2** and **1**, respectively). Since the orientations of the surrounding P-Ph rings around the H-2 of **1** and **2** are similar (Fig. 1c, d), the shielding effects caused by the P-Ph rings should be comparable. Therefore, the above difference may reflect the strength of the Au···H–C interactions. Thus the Au···H–C interactions in **2** seem weaker than in **1**, in good agreement with the difference in the Au–H distances in the

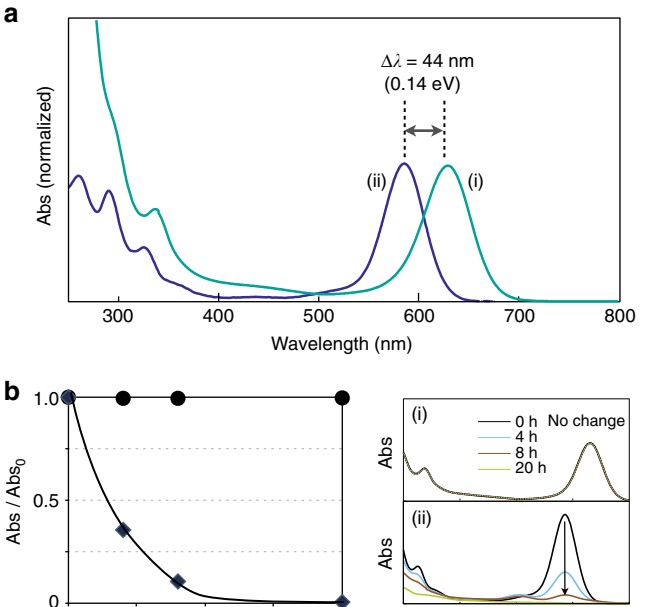

**Fig. 3** Absorption spectra and stability tests. **a** Electronic absorption spectra of (i) **1**·(PF$_6$)$_2$ and (ii) **2**·(NO$_3$)$_2$ in CH$_2$Cl$_2$ at 25 °C with visible absorption maxima normalized. **b** Monitoring of absorption spectra of (i) **1**·(PF$_6$)$_2$ and (ii) **2**·(NO$_3$)$_2$ in CH$_2$Cl$_2$ under ambient light and temperature (~25 °C) and time courses of Abs/Abs$_0$ at 630 nm of **1**·(PF$_6$)$_2$ (filled circles) at 593 nm of **2**·(NO$_3$)$_2$ (filled diamonds)

crystal structures (Table 1). These differences between **1** and **2** may be correlated to the difference in the intrinsic donor strength of C($sp^2$)-H and C($sp^3$)-H groups[36].

**Absorption spectra and stability**. As noted in the introductory part, the apparent colours of **1**·(PF$_6$)$_2$ and **2**·(NO$_3$)$_2$ were markedly different; **1**·(PF$_6$)$_2$ was greenish blue while **2**·(NO$_3$)$_2$ deep blue. Accordingly, in the electronic absorption spectra (Fig. 3a), **1**·(PF$_6$)$_2$ showed an intense isolated band at 631 nm, which was shifted by 45 nm from the band of **2**·(NO$_3$)$_2$ (586 nm). No effects of the counter anions (PF$_6^-$ vs. NO$_3^-$) were observed in the visible absorption profiles. A previous theoretical study on the electronic structure of **2** showed that the visible absorption is virtually due to the metal-to-metal HOMO–LUMO transition within the Au$_6$ skeleton in which the 6sp orbitals are primarily involved[31]. Consequently, when two Au$_6$ clusters have almost identical skeletal structures, as in the case of **1** and **2** (Fig. 1c, d), they should give similar absorption energies. However, the band positions were marked different. This observation implies the electronic perturbation effects of the neighbouring π-electrons of the phenylene bridges on the Au$_6$ moiety of **1**[37, 38], which may be correlated to its strong Au···H–C interactions. Although at present the possibility of through-space electronic coupling cannot be excluded, the above results may suggest interesting opportunities to tune the electronic structures and optical properties of gold clusters by proximal organic chromophore through non-covalent interactions. Theoretical studies on the role of Au···H–C interactions are worthy of future investigations.

Finally, we would like to point out that the strength of the Au···H–C interactions critically affects the cluster stability. The Au$_6$ framework, which is basically held by weak Au–Au bonding together with the chelating diphosphine ligands, is known to undergo decomposition or transformation to different cluster

species under particular conditions. For example, the trimethylene-bridged cluster **2**·(NO$_3$)$_2$ in dichloromethane gradually lost its blue colour under ambient light and temperature. The absorption bands at 586 and ~ 320 nm were decreased in intensity, and almost faded away to give a colourless solution after 24 h (Fig. 3b, *right*). In contrast, such an absorption (colour) loss was hardly observed for **1**·(PF$_6$)$_2$. Even at 80 °C in *o*-dichlorobenzene, **1**·(PF$_6$)$_2$ was stable to give the original spectral pattern and absorbance after 2 h, while **2**·(NO$_3$)$_2$ was instantly decomposed within 1 min under the similar conditions. These behaviours should reflect the effective stabilization of the Au$_6$ framework of **1** through the strong Au···H–C interactions, as found in the X-Ray structures and NMR profiles, presumably assisted by the above-mentioned cluster-π interaction.

## Discussion

We have demonstrated that a cationic four-electron Au$_6$ cluster ([Au$_6$]$^{2+}$) can interact with proximal unfunctionalized C–H. Albeit cautions must be taken for interpretation of the shifts of the NMR signals, the downfield shifts observed here are fairly large ($\Delta\delta = > 4$ and $> 10$ ppm for $^1$H and $^{13}$C, respectively) and may not be explained only by the factors other than the electrostatic-based forces. According to the latest definition recommended by the IUPAC task team[39], "The hydrogen bond is an attractive interaction between a hydrogen atom from a molecule or a molecular fragment X–H in which X is more electronegative than H, and an atom or a group of atoms in the same or a different molecule, in which there is evidence of bond formation." Based on this criterion, it can be reasonably concluded that the present Au···H–C interaction is a kind of "hydrogen bond", where the [Au$_6$]$^{2+}$ serves as an acceptor. To our knowledge there have been no examples of Au···H hydrogen bonds spectroscopically identified, but some theoretical studies predicted the possibility of gold clusters to form hydrogen bonds with H–X fragments[40–43]. In this respect, this work shows the above predictions are essentially true and discloses the unique activity of the electrons delocalised over the gold units. Thus gold cluster compounds having partially oxidised metallic units are virtually different from the conventional complexes with single metal centres. The capability to interact with unfunctionalized C–H groups via non-covalent forces may not only promote the further understanding of chemical bonding but also shed light on the elucidation/developments of recently emerging gold cluster catalysis[27–30].

## Methods

**Materials and methods**. 1,3-Bis(diphenylphosphino)propane (TMDP, 97%) was purchased from Aldrich. 1,3-Bis(diphenylphosphino)benzene (*m*PhDP) was synthesized by the reaction of *m*-dibromobenzene with *n*-butyllithium (2 equiv) in diethyl ether (−78 °C then reflux for 3 h) followed by the addition of chlor-odiphenylphosphine (2 equiv) (room temperature, 24 h)[44]. [Au$_9$(PPh$_3$)$_8$](NO$_3$)$_3$ was prepared by the NaBH$_4$ reduction of Au(PPh$_3$)(NO$_3$) in ethanol[45] and purified by recrystallisation by vapor diffusion of diethyl ether into a methanol solution. [Au$_6$(TMDP)$_4$](NO$_3$)$_2$ (**2**·(NO$_3$)$_2$) was prepared by the reaction of [Au$_9$(PPh$_3$)$_8$](NO$_3$)$_3$ with 1,3-bis(diphenylphosphino)propane (TMDP) and purified by recrystallisation by vapor diffusion of ether into a solution of the cluster in dichloromethane[33]. The other reagents and solvents are given in Supplementary Methods. $^1$H-NMR, $^1$H-$^1$H DQF COSY NMR, and $^{31}$P-NMR were measured on a JEOL EX-400 or a JEOL JNM-ECA600 NMR spectrometer. $^{13}$C NMR and HSQC spectra were obtained on a JEOL JNM-ECA600 NMR spectrometer at Faculty of Science, NMR laboratory, Hokkaido University. $^1$H and $^{13}$C NMR chemical shifts (in ppm) were referenced to internal CDHCl$_2$ (5.32 and 53.82 ppm, respectively). $^{31}$P-NMR chemical shift (in ppm) was referenced to 85% H$_3$PO$_4$. ESI-MS spectrum was recorded on a Bruker micrOTOF-HS in methanol.

**Synthesis of [Au$_6$(*m*-PhDP)$_4$](PF$_6$)$_2$ (1·(PF$_6$)$_2$)**. To a dichloromethane solution (4 ml) of [Au$_9$(PPh$_3$)$_8$](NO$_3$)$_3$ (50.0 mg) in a 50 ml flask was added 1,3-Bis(diphenylphosphino)benzene (*m*PhDP) (20 eq., 110.1 mg in 3 ml dichloromethane) drop-wise. The color of the solution turned from deep blue to intense green during

the addition. After the mixture was stirred at r.t. for 90 min in the dark, the solvent was evaporated and the residue was dissolved in dichloromethane (4 ml). The solution was poured into diethyl ether (150 mL), and the resulting precipitate collected by filtration was washed with diethyl ether and hexane to give $1 \cdot (NO_3)_2$ as greenish blue solids (47.0 mg, quant.). [1]H-NMR ($CD_2Cl_2$): δ 6.49 (q, 16H), 6.68 (t, 16H), 6.89-6.99 (m, 20H), 7.06-7.13 (m, 24H), 7.38 (t, 12H), 7.52 (t, 4H), 11.58 (m, 4H). [31]P-NMR ($CD_2Cl_2$): δ 61.36, 52.10. $1 \cdot (NO_3)_2$ thus obtained above was dissolved in methanol (5 ml) and $KPF_6$ (excess) was added. After 20 min, solvent was evaporated and the residue was dissolved in dichloromethane (4 ml). Diethyl ether (100 ml) was added to the solution. The precipitate filtered was washed with diethyl ether to give $1 \cdot (PF_6)_2$ (19.0 mg, 47%) as green solids. Crystals suitable for X-ray analysis were obtained by vapor diffusion of diethyl ether into a solution of dichloromethane of the cluster. Elemental analysis calcd (%) for $Au_6(C_{30}H_{24}P_2)_4 \cdot (PF_6)_2 \cdot (CH_2Cl_2)_2$ ($C_{122}F_{12}H_{100}P_{10}Au_6Cl_4$): C 42.75 H 2.94; found: C 42.95 H 2.95. [1]H-NMR ($CD_2Cl_2$): δ 6.49 (q, 16 H), 6.68 (t, 16 H), 6.89-6.98 (m, 16 H), 6.97 (br, 4 H), 7.06-7.13 (m, 24 H), 7.38 (t, 12 H), 7.52 (t, 4 H), 11.57 (m, 4 H). [31]P NMR ($CD_2Cl_2$): δ 61.36, 52.09, −135.68.

**Synthesis of digold(I) complexes (3 and 4).** 3 and 4 were prepared according to the standard procedure using the reaction of phosphine and gold(I) dimethylsulfide complex[46]. $Au_2(mPhDP)Cl_2$: [1]H-NMR ($CD_2Cl_2$): δ 7.20 (m, 1H), 7.43-7.49 (m, 16H), 7.50-7.57 (m, 4H), 7.64 (m, 1H), 7.82 (m, 2H). [31]P NMR ($CD_2Cl_2$): δ 33.21. $Au_2(TMDP)Cl_2$ was prepared similarly. [1]H-NMR ($CD_2Cl_2$): δ 1.92 (m, 2H), 2.79 (m, 4H), 7.48-7.50 (m, 12H), 7.63-7.68 (m, 8H). [31]P NMR ($CD_2Cl_2$): δ 26.61.

**Crystallography.** Crystal data were collected on a Bruker SMART Apex II CCD diffractometer with graphite-monochromated MoKα radiation ($\lambda = 0.71073$ Å). The crystal structure was solved by direct methods (SHELXS-2013) and refined by full-matrix least-squares methods on $F^2$ (SHELXL-2014) with APEX II software. Non-hydrogen atoms were refined anisotropically. Hydrogen atoms were located at calculated positions (riding model) and refined isotropically. The structure of $2 \cdot (NO_3)_2$ was reported in 1982 by van der Velden et al.[33], and is available in The Cambridge Structural Database (CSD version 5.37) (Refcode BOTSOS). However, the data quality is not satisfactory (no H atom information is given), so we newly prepared the crystals and carried out the structural analyses. Detailed parameters are given in Supplementary Tables 1, 2.

**Data availability.** The X-ray crystallographic data have been deposited in the Cambridge Crystallographic Data Centre with CCDC-1541477 ($1 \cdot (PF_6)_2$) and CCDC-1541478 ($2 \cdot (NO_3)_2$).

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

## Acknowledgements

This work was partly supported by MEXT/JSPS Grant- in-Aid (16H05961), Iketani Science and Technology Foundation, and Mitsubishi Foundation. We thank Dr. Yasuhiro Kumaki, Hokkaido University, for the measurement of high-resolution NMR spectra.

## Author contributions

K.K. supervised and guided the research, summarized data, and wrote the manuscript. Md.A.B. performed the synthesis and routine spectral measurements of the mPhDP ligand and **1**, and contributed to the preparation of the first draft of the manuscript. M.I. contributed NMR measurements and data analyses. M.S. and Y.S. performed crystallographic analyses. All of the authors read the manuscript.

## Additional information

**Competing interests:** The authors declare no competing financial interests.

