## [Peer Review File · Nature Communications]

Editorial Note: Parts of this peer review file have been redacted as indicated to maintain the confidentiality of unpublished data

Reviewers' comments:

Reviewer #1 (Remarks to the Author):

This is a somewhat interesting paper in which spectroscopic data are used to propose a hydrogen bond from a CH fragment to gold, Au...HC. The problem I see is that there are no theoretical calculations to help better understand the details of the interaction.

There are other papers on Au...HX clusters, both experimental and theoretical, so novelty here is not that great (see Kryachko JPCA 2005; Tehrani JPCA 2012).

The chemical shift effect is there but the shift is not that great, the delta delta 4 ppm shift change and 12 ppm isotropic chemical shifts are much smaller than seen in strongly H-bonded systems. Calculations might help quantitate the shifts seen and put some numbers on the bond energies, though Au could be a challenge.

So the paper is interesting but not that exciting.

Reviewer #2 (Remarks to the Author):

The detailed crystallographic and spectroscopic studies of related gold cluster complexes reported in the paper have demonstrated to my satisfaction the first example of spectroscopically identified hydrogen bonding interaction of C-H units to Au atoms in divalent hexagold cluster ([Au₆]²⁺ decorated by diphosphine ligands). Their X-ray crystallographic studies were hampered because of the well documented fact that the X-rays interact with the electrons in the C-H bonds rather than the nuclei. The authors have made appropriate corrections which reveal significantly shortened Au-H / Au-C distances and strongly indicate the presence of attractive interactions involving unfunctionalized C-H moieties and the gold atoms. Solution ¹H and ¹³C NMR signals of the C-H units appeared at considerably downfield regions, indicating the hydrogen-bond character of the interactions. These interactions are not very strong, but sufficiently significant for them to be reported and documented. These interactions also influence the electronic spectral characteristics of the clusters and perhaps may have an important stabilising effect. This work demonstrates the uniqueness and potential of partially oxidised Au cluster moieties to participate in non-covalent interaction with various organic functionalities. Clearly future experimental work using neutron diffraction studies and theoretical studies will provide more information regarding such interactions. However, I think that the work in its present form is sufficiently important and interesting to merit publication in this journal.

Reviewer #3 (Remarks to the Author):

Comments to Authors

Potential hydrogen bond formation to anionic and neutral gold clusters have previously been studied by chemical computation. The present manuscript presents the first evidence that such anagostic (in the Brookhart and Green's sense (Ref. 1)) interactions are possible with positively charged clusters, albeit when this charge is partly neutralised by σ -donating ligands. Furthermore, these chosen ligands position active hydrogen atoms in close proximity of one or more metal atoms. Other known criteria for describing the interaction as hydrogen bond formation, are fulfilled by convincing experimentally determined parameters.

The results described should be of interest to both theoretical and experimental chemists and could also be useful to researchers in photophysics. Moreover, the manuscript describes an important step towards an appreciation of often claimed weak gold-hydrogen interactions. Future attention should be given in particular to the role played by the ligands in electronically preparing the positively charged cluster for electrostatic interaction with C-bonded hydrogen atoms.

As regards further strengthening of the manuscript, the authors should preferably include J_{C-H} coupling constants of the interacting C–H units and compare them with the other relevant C–H's. It is known that the coupling constants are less important for anagostic compared to agostic interaction but the data here are lacking. In addition, exchanging the counterions of **1** and **2** and comparing NMR results for the new products with the existing ones, should give a much better idea of the role played by these anions. Finally, are these complexes 1:2 electrolytes in CH₂Cl₂ solutions?

The manuscript is clearly written but careful attention should be given to the following:

1. The language needs editing e.g.

p. 2, line 4: ...is the agostic bond ...

p. 2, second par: ...a special position...

p. 3, line 6 from bottom: ...a trimethylene-bridged...

p. 5, line 3 from bottom: The previous sentence is repeated.

p. 9, last par.: Do not repeat what has been said before.

2. A recent article by Esterhuysen et al. (*Angew. Chem. Int. Ed.* **2016**, 55, 1694) discussed the very important role played by ligands during AuI...H interactions, and could be useful to readers.

3. Fig. 1a and 1b on p. 5 should be replaced by **1c** and **1d**.

4. The sum of van der Waals radii (p. 6) are reported as 2.86 and 2.88 Å, which is correct?

5. p 6, line 2: Fig. **1e-g**: use bold letters

6. It could help the reader if the 'the reference complex' (p. 7 and 9) is numbered **3**.

7. Table 2: dinuclear complex, **3**.

8. p. 9, line 12 : 1c should be **1e**.

9. p. 10: 'Intrinsic donor strength' is not measured by pKa-values. Gas phase proton affinities could give a better indication.

10. The origin of the colour changes mentioned on p. 3 needs further investigation before the H...Au interaction could be involved.

11. Abstract: For Au-I there are decisive evidence. Explain the last sentence.

12. Consider changing the title toAu atoms in coordinated gold clusters.

13. References:

i. Make sure et al. (if allowed) is used consistently (|Ref 28?)

ii. Ref. 20:Au...H–X

iii. Ref. 40: Abbreviation?

iv. Ref 41: Show pages m- 1 – ?

I enjoyed reading this manuscript.

Responses to the comments of Reviewer# 1.

Redacted

Redacted

There are other papers on Au...HX clusters, both experimental and theoretical, so novelty here is not that great (see Kryachko JPCA 2005; Tehrani JPCA 2012).

Reply: To our knowledge, there have been very few examples of experimentally evidenced Au...H hydrogen bonds. The two references are theoretical-only papers and do not offer any experimental results.

The chemical shift effect is there but the shift is not that great, the delta delta 4 ppm shift change and 12 ppm isotropic chemical shifts are much smaller than seen in strongly H-bonded systems. Calculations might help quantitate the shifts seen and put some numbers on the bond energies, though Au could be a challenge.

Reply: We think the observed ^1H NMR downfield shift (more than 4 ppm) is fairly large with a normal sense of experimental chemists, especially in organic chemistry, since the reported shifts in strong intramolecular $\text{NH}\cdots\text{O}$ and $\text{NH}\cdots\text{F}$ systems are at most ~ 3 ppm. Furthermore the chemical shift of ~ 12 ppm is quite unusual for aromatic C-H protons, considering they generally appear at 6.5 – 8.5 ppm. The comparison with very strong systems seems interesting but we think it is out of focus of this paper. Rather, we would like to emphasize that Au...H hydrogen bonds definitively exist in the real world, even if they are not so strong.

Responses to the comments of Reviewer# 3.

As regards further strengthening of the manuscript, the authors should preferably include J_{C-H} coupling constants of the interacting C–H units and compare them with the other relevant C–H's. It is known that the coupling constants are less important for anagostic compared to agostic interaction but the data here are lacking.

Reply: The J_{CH} value and an associated sentence were added (p. 9, lines 128-132).

*In addition, exchanging the counterions of **1** and **2** and comparing NMR results for the new products with the existing ones, should give a much better idea of the role played by these anions. Finally, are these complexes 1:2 electrolytes in CH₂Cl₂ solutions?*

Reply: The NMR spectra of **1**·(NO₃)₂ and **1**·(PF₆)₂ were almost similar to each other, indicating that the effect of counter ion is negligible and they basically exist as electrolytes in CH₂Cl₂. In the revised manuscript, we added the result (p. 7, lines 112-114) and overlaid NMR spectra of **1**·(NO₃)₂ and **1**·(PF₆)₂ in Figure 6 of supplementary information.

The manuscript is clearly written but careful attention should be given to the following:

1. The language needs editing e.g.

p. 2, line 4:is the agostic bond

p. 2, second par:a special position....

p. 3, line 6 from bottom:a trimethylene-bridged....

p. 5, line 3 from bottom: The previous sentence is repeated.

p. 9, last par.: Do not repeat what has been said before.

Reply: These were corrected taking account of the suggestions.

*2. A recent article by Esterhuysen et al. (Angew. Chem. Int. Ed. **2016**, 55, 1694) discussed the very important role played by ligands during AuI···H interactions, and could be useful to readers.*

Reply: The suggested paper, which reports a theoretical study on Au···H hydrogen bonds, was cited (ref. 43).

*3. Fig. 1a and 1b on p. 5 should be replaced by **1c** and **1d**.*

4. The sum of van der Waals radii (p. 6) are reported as 2.86 and 2.88 Å, which is correct?

*5. p 6, line 2: Fig. **1e-g**: use bold letters*

*8. p. 9, line 12 : 1c should be **1e**.*

Reply: Mistyping were corrected as carefully as possible. For 4, 2.86 Å is correct (p. 6, line 3 from the bottom).

*6. It could help the reader if the 'the reference complex' (p. 7 and 9) is numbered **3**.*

*7. Table 2: dinuclear complex, **3**.*

Reply: The reference dinuclear complexes were numbered **3** and **4** (p. 7, 8 and 10).

9. p. 10: 'Intrinsic donor strength' is not measured by pKa-values. Gas phase proton affinities could give a better indication.

Reply: The description of the pKa values in the original manuscript was deleted. Instead the difference of intrinsic donor strengths was noted with a reference (p. 10, lines 161-162).

10. The origin of the colour changes mentioned on p. 3 needs further investigation before the H...Au interaction could be involved.

Reply: As noted in the text (p. 10, last paragraph), the colour difference between **1** and **2** (absorption spectral difference) should be associated with the electronic perturbation effects of the π -electrons of the phenylene bridge of **1**, since the geometries of the Au₆ framework are almost identical. For molecular-level elucidation for this observation, structural information, such as the orientation of the phenylene ring, would be essential, so we first attempted the structural analyses (NMR and X-ray). DFT and TD-DFT studies are in progress based on the crystal structure.

11. Abstract: For Au-I there are decisive evidence. Explain the last sentence.

Reply: We understand that the reviewer points out the case of auride anion complex ([Au]⁻), for which there exists an example demonstrating the presence of Au⁻...H interaction in the crystal structure. To avoid confusion of readers, the corresponding part (p. 1, line 11) was revised as follows.

...there have been no decisive evidences... was replaced with
...there have been limited decisive experimental evidences...

12. Consider changing the title toAu atoms in coordinated gold clusters.

Reply: We appreciate the reviewer's suggestion, but would like to keep the original title because this work offers the first experimental-based example of Au...H hydrogen bonds found in the gold cluster family, including bare clusters.

13. References:

- i. Make sure et al. (if allowed) is used consistently (|Ref 28?)
- ii. Ref. 20:Au...H-X
- iii. Ref. 40: Abbreviation?
- iv. Ref 41: Show pages m- 1 – ?

Reply: Mistyping in the reference section were corrected and checked as carefully as possible.

REVIEWERS' COMMENTS:

Reviewer #3 (Remarks to the Author):

My recommendation regarding a title change was not accepted by the authors. Yet, it is highly unlikely that a naked gold cluster carrying positive charge would interact with a proton donor and, therefore, pertinently important to acknowledge in the paper the determining role played by the phosphine ligands in lowering the partly positive charge on the gold atoms.

Responses to the comments of Reviewer# 3.

My recommendation regarding a title change was not accepted by the authors. Yet, it is highly unlikely that a naked gold cluster carrying positive charge would interact with a proton donor and, therefore, pertinently important to acknowledge in the paper the determining role played by the phosphine ligands in lowering the partly positive charge on the gold atoms.

Reply: We now understand the intention of the reviewer. The electron donation from the P-atoms would certainly neutralize the positive charge of the Au₆ moieties which may be critical to facilitate the interaction with weakly polar C-H groups. We are glad to accept the recommendation to change the paper title from "...in gold clusters" to "...in coordinated gold clusters", according to the previous comment.